# Evaluation of a new transpalpebral tonometer for self-measuring intraocular pressure

Hidenaga Kobashi [1,2]*

1 Toneasy Inc., Tokyo, Japan, 2 Department of Ophthalmology, School of Medicine, Keio University, Tokyo, Japan

* himon@hotmail.co.jp

## Abstract

### Purpose

We developed a novel transpalpebral self-tonometer called the TapEye tonometer (TET) based on palpation of the upper eyelid. Our goal was to evaluate a method for improving the accuracy of measuring intraocular pressure (IOP) through the eyelid.

### Methods

Participants underwent standardized training by technicians and were required to be able to use the TET for study inclusion. Subsequently, a noncontact tonometer and Goldmann applanation tonometer (GAT) were used. All participants were instructed to measure their IOPs using the three tonometers at baseline (visit 1) and at 1 month (visit 2). At visit 2, the corrected IOP value measured by the TET (c-TET) was calculated using the difference between the TET and GAT measurements obtained at visit 1.

### Results

No significant correlations were found between the TET and GAT measurements at any visit, but the correlation between the c-TET and GAT measurements at visit 2 was significant. The mean difference between the c-TET and GAT measurements was 0.4 ± 3.7 mmHg in the right eye and 0.5 ± 3.4 mmHg in the left eye.

### Conclusions

After correcting the IOP based on the difference between the TET and GAT measurements at the initial visit, the corrected IOP value of the TET was correlated with that of the GAT at the second visit. The TET has the potential to address an unmet need by providing a tool for minimally invasive IOP measurements.

### Trial registration

**Clinical trial registration number:** jRCTs032220268.

**Data Availability Statement:** All relevant data are within the paper.

**Funding:** This is based on results obtained from a project, JPNP0407002, subsidized by the New Energy and Industrial Technology Development Organization (NEDO). The funders had no role in

study design, data collection and analysis, decision to publish, or preparation of the manuscript. The authors received no specific funding for this work. The author obtained the funding as JPY 24,770,681 in this grant by NEDO.

**Competing interests:** H.K.: CEO and equity owner, Toneasy Inc.; Patent, Toneasy Inc. This does not alter our adherence to PLOS ONE policies on sharing data and materials.

**Abbreviations:** IOP, intraocular pressure; CCT, central corneal thickness; NCT, noncontact tonometer; GAT, Goldmann applanation tonometer; TET, TapEye tonometer; c-TET, corrected-TapEye tonometer; DUES, deepening of the upper eyelid sulcus.

## Introduction

Glaucoma is one of the leading causes of legal blindness worldwide. An elevated intraocular pressure (IOP) is recognized as the most important and only controllable risk factor leading to progressive glaucomatous damage [1]. All treatments for glaucoma aim to lower IOP; therefore, accurate tonometry is crucial for diagnosis and treatment evaluation. The Goldmann applanation tonometer (GAT) is considered the current gold standard method for measuring IOP [2]. However, this technique has various drawbacks; specifically, it requires the use of topical anesthesia and has difficulty in accurately measuring IOP in children and/or physically handicapped patients. Generally, IOP measurement during a clinical consultation provides a reading that is not reflective of the IOP at any other point in time. Often, the ophthalmologist must make clinical decisions such as setting target IOPs and assessing treatment responses based on the assumption that the IOP remains fairly constant throughout the day. Moreover, the range of diurnal variation in IOP in patients with glaucoma has been found to be 2 to 3 times that of normal individuals [3]. Greater fluctuations in IOP are associated with later stages of glaucoma [4–7].

With the advent of self-tonometry, 24-hour monitoring at home may become a viable option. The ideal home tonometer would be a reliable, user-friendly, portable, and convenient device. Although various forms of ocular tonometry exist, rebound tonometry is currently a popular choice [8]. The Icare Home (TA022, Icare Oy, Vanda, Finland) is a relatively new rebound tonometer; approved by the US Food and Drug Administration in March 2017, it is specifically designed for self-measuring IOP at home. This device is capable of recording out-of-office variations in IOP through self-tonometry and provides electronic documentation of these measurements. Subsequently, transpalpebral tonometers such as TGDc-01, Diaton [9], and pressure phosphene tonometers [10], portable devices that measure IOP through the eyelid, were developed. However, these transpalpebral tonometers have not been widely used for self-measurement because of their unacceptable accuracy, repeatability, usability, and branding. Here, we developed a novel transpalpebral self-tonometer called the TapEye tonometer (TET), which is based on palpation of the upper eyelid. This palpation method has been used for the estimation of IOP for many years and is a popular choice for postkeratoplasty and postcataract surgery patients in a clinical setting. The purpose of this proof-of-concept study of the TET is twofold: to evaluate the accuracy of the measured IOP compared with that measured with a Goldmann applanation tonometer (GAT) and to analyze the differences in the measurements between the two tonometers.

## Methods

### 3D-printed eye model study

To verify the measurement of IOP using the TET, we generated an elevated IOP model with 3D printing technology [11]. Eight eyes with different central corneal thicknesses were prepared to simulate different IOPs. With the 3D-printed eye model, we confirmed that corneal thickness was significantly correlated with IOP using a Tono-Pen AVIA tonometer (Reichert Inc., Depew, New York, USA) [11]. To simplify the simulation of the elevated IOP model, we generated artificial eye models without eyelids by changing the corneal thickness via 3D printing. The TET tip was placed close to the 3D-printed eye model without touching it and parallel to the ocular apex center. The push button of the TET was vertically pressed against the model eye by the same technician using their finger (H.K.). Five successful measurements were recorded at each corneal thickness level, and the mean value and standard deviation of each measurement were recorded. The TET software determined the optimal parameter thresholds,

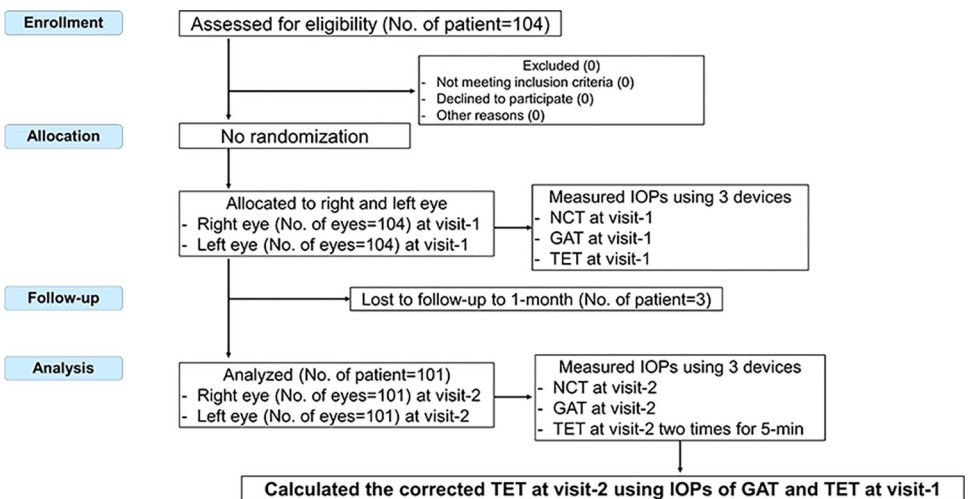

**Fig 1. CONSORT flowchart for the single-arm, open-label, clinical trial with the TapEye tonometer in normal and glaucoma subjects.** IOP = intraocular pressure; NCT = noncontact tonometer; GAT = Goldmann applanation tonometer; TET = TapEye tonometer.

such as acceleration, speed, and alignment, for pressing the eyeball, but the details of this program are a trade secret.

## Clinical study

This prospective, single-center, nonrandomized trial assessed the accuracy and repeatability of the TET tonometer. Institutional review board approval was obtained, and the study subjects provided written informed consent to participate. The study adhered to the tenets of the Declaration of Helsinki. This trial was approved by the Certified Review Board at Hattori Clinic and registered in the Japan Registry of Clinical Trials (jRCT): jRCTs032220268. Our study followed the CONSORT recommendations; the flowchart is shown in **Fig 1**. Participants were recruited from Yokohama Tsurumi Chuo Eye Clinic, Kanagawa, Japan, between October 2022 and February 2023. All subjects underwent a slit lamp examination, including fundoscopy without pupillary dilation, by an experienced ophthalmologist (T.T.) before the IOP measurements were obtained.

## Inclusion and exclusion criteria

Male or female individuals of any race or ethnicity and aged 18 years or older were eligible to participate. If the subjects had been previously diagnosed with glaucoma, they were able to participate in this study and did not have to stop taking glaucoma drops. Contact lens wearers were required to remove the contact lenses before the IOP measurements. All participants were also required to visit the clinic at two time points, at baseline (visit 1) and 1 month (visit 2).

The exclusion criteria included a history of ocular surgery 1 month before the initial visit. Subjects who underwent clinical ophthalmic consultation or/and surgery during the research period were also excluded. Patients with a history of incisional glaucoma surgery, LASIK, or corneal surgery were not excluded, nor were patients who had undergone cataract extraction when one month or more passed. We did not exclude patients with a history of previously diagnosed keratoconus.

## The TapEye tonometer

The development of the TET began in 2018 when the author (H.K.) and another engineer submitted it to the patent office. The author started Toneasy, Inc., in 2020 and made a prototype TET. This study used the latest prototype, which taps on the upper eyelid for self-measurement. It allows for noninvasive measurement of IOP and permits transpalpebral measurement without contact with the cornea or the need for local anesthesia. This tonometer is now an off-label medical device in Japan in 2023. The components of the device are shown in **Fig 2**. The TET is divided into three parts: a body component and two support components. The two support plates are attached to the forehead and cheek and can be removed by sliding and repositioned to perform measurement on the other eye. To fit individual face shapes, the height of the support plate can be adjusted using dial screws.

The self-measurement procedure with the TET is described below (**Fig 3** **and S1 Video**):

- The device is held in the nondominant hand, and the button is pushed by a finger from the dominant hand with the subject in a sitting position.

- The subject places the tonometer tip onto their eyelid, approximately 1 to 3 mm behind the lid margin of the upper eyelid; the tip location corresponds to the corneal center.

- The strength, speed, acceleration, distance, and misalignment of the pushing stylus are measured by the sensor and determined by a computer with installed software. The threshold for each criterion is confidential.

- If the IOP data pass the criteria, the device chimes to indicate success. If they do not pass, the device rings an alarm to indicate failure.

- Subjects continue to push the button until seven successful have been achieved, after which the software provides the calculated data. The average value of the seven measurements is

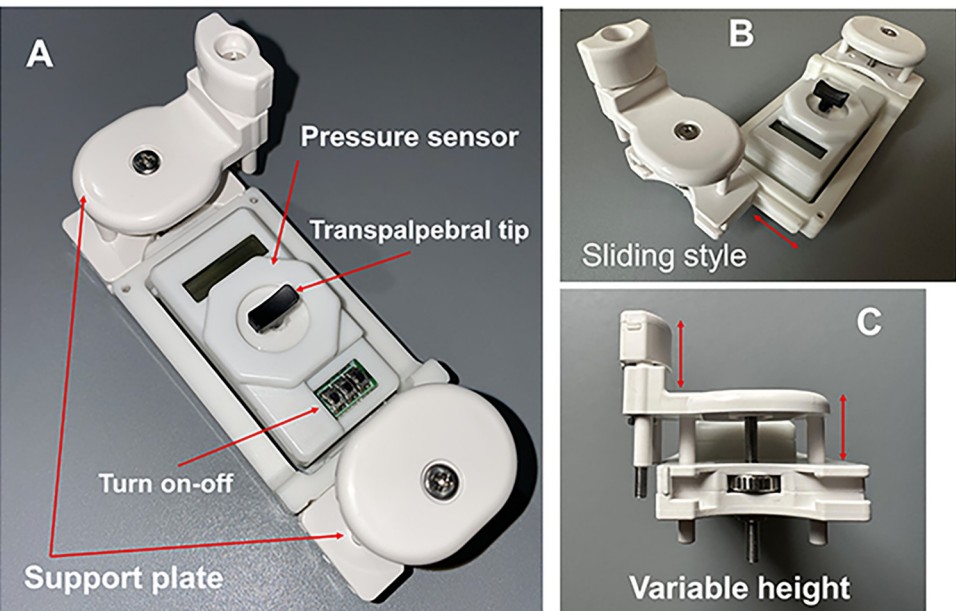

**Fig 2. The TapEye tonometer.** A, Overall appearance. B, The support plate is removed by sliding and repositioned onto the other eye. C, The height of the support plate can be adjusted to accommodate various face shapes.

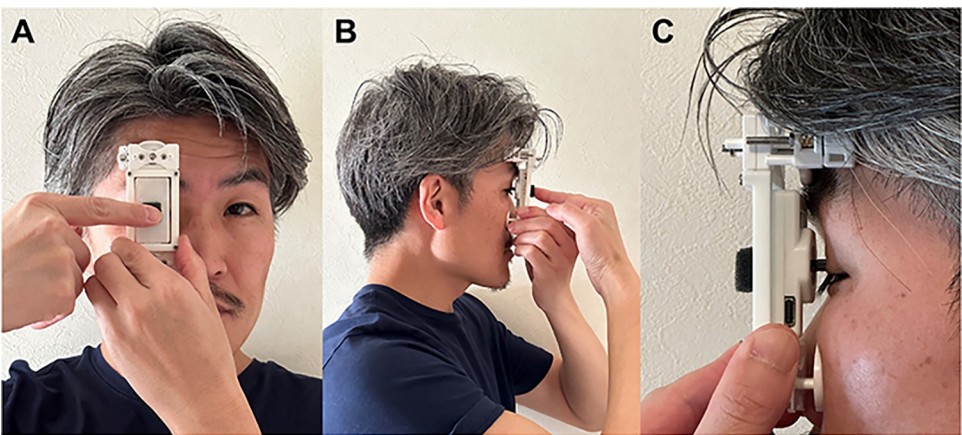

**Fig 3. The appearance of the TapEye tonometer during the measurement.** A, Front view: tapping the button of the device with a finger. B, Lateral view. C, Magnified image of the tip and eyelid. The upper eyelid is naturally closed. The subject places the tonometer tip onto their eyelid, approximately 1 to 3 mm behind the lid margin of the upper eyelid; the tip position corresponds to the corneal center.

considered the IOP value measured by the TET. Failed measurements are excluded from the IOP calculation.

A calibration check was performed before the study visit.

## IOP measurements at visit 1

Enrolled subjects watched an instructional video for using this device before taking any measurements. They underwent standardized training (lasting approximately 10 to 15 minutes) in the use of the TET, followed by a certification procedure involving noncontact tonometer (NCT) and GAT measurements. A TET tonometer was available to each individual subject, and the supporting parts of the device were customized to appropriately fit to the eyelid by technicians (H.K., K.S., K.T.). While in a seated position, the subjects held the device in their nondominant hand and pushed the button using a finger from their dominant hand with appropriate strength, speed and acceleration (**Fig 3**). Technicians observed the subjects using the device. When the subjects asked for help or held the device in an inappropriate manner and pushed the button, the technicians gave the proper instructions while standing by their side at both visits. The technicians were unaware of the NCT and GAT data, as the TET measurement procedure was performed first. Based on the outcomes confirmed to be successful by the TET, the subjects continued to measure their IOP, and then the data were automatically recorded by a computer with installed software.

Approximately 5 minutes after these measurements were taken, an NCT measurement was performed by a technician (A.K.) who was blinded to the TET and GAT measurements. The NCT measurements were take by a recently developed device, the Tono-Pachymeter NT-530P device (Nidek Co., Ltd., Gamagori, Japan), which simultaneously performs central corneal thickness measurements as well. Subsequently, GAT measurements were performed by the same ophthalmologist (T.T.), who was not aware of the TET and NCT measurements. With the GAT, only a single measurement was performed at each visit.

## IOP measurements at visit 2

One month after visit 1, the same IOP measurements were performed using the three tonometers at visit 2. The initial instruction and self-training procedures were omitted. To evaluate

the measurement repeatability of the TET, the second measurement was performed 5 minutes after the first measurement.

At visit 2, the corrected IOP value of the TET (c-TET) was derived from the difference between the TET and GAT measurements (Δ IOP) at visit 1. The formula for calculating c-TET was initially suggested as follows:

$$c-TET = TET_{visit-2} - \Delta IOP$$

where

$$\Delta IOP = TET_{visit-1} - GAT_{visit-1}$$

## Safety outcomes

At visits 1 and 2, a slit-lamp examination was performed to evaluate adverse events, such as abrasions, eyelid disorders, and conjunctivitis.

## Statistical analysis

Analyses were performed with Statistical Analysis Software (version 9.4; SAS Institute, Cary, NC). The outcome measures are reported as the mean ± standard deviation (SD). The Kolmogorov–Smirnov test was used to assess the normality of the distributions. Bonferroni-adjusted parametric tests (analysis of variance (ANOVA)) were applied to test for significant differences between devices. For normally distributed data, correlations were evaluated with Pearson's r coefficient. For nonnormally distributed data, Spearman's r was calculated. To confirm the consistency in the IOP between the NCT and GAT, we performed a statistical correlation analysis.

Bland–Altman analysis was used to assess the bias and 95% limits of agreement between c-TET and GAT. In the Bland–Altman analysis, the difference between each IOP measurement was plotted against the mean. Linear regression of the Bland–Altman analysis showed whether any over- or underestimation of IOP had occurred within the measured range. To clarify the consistency of the Bland–Altman analysis, we analyzed the correlation plots between the values of c-TET-GAT and GAT at visit 2. In this study, repeatability was determined using the intra-class correlation coefficient (ICC) and coefficient of variation (CV) values. A two-tailed paired t test was used in the statistical analyses to compare the elastic modulus. A p value of $< 0.05$ was considered to indicate statistical significance.

## Results

### 3D-printed eye model study

We successfully measured the IOPs of several 3D-printed eyeball models with various central corneal thicknesses using the TET and Tono-Pen AVIA tonometer. Correlation analysis indicated a strong and significant correlation between the TET and Tono-Pen AVIA IOP measurements of the 3D-printed eyeball models (**Fig 4**).

### Clinical study

One hundred four subjects were recruited and agreed to participate in the study. At visit 1, all participants completed the measurements using the three tonometers. Three subjects failed to complete a 1-month follow-up due to scheduling conflicts. The remaining 101 subjects completed the study measurements at visit 2 and were included in the calculation of c-TET and the repeatability analysis (**Fig 1**). The demographics of the study participants are summarized in **Table 1**. The participants had a mean age of 40.3 years (range: 20 to 87 years). Most

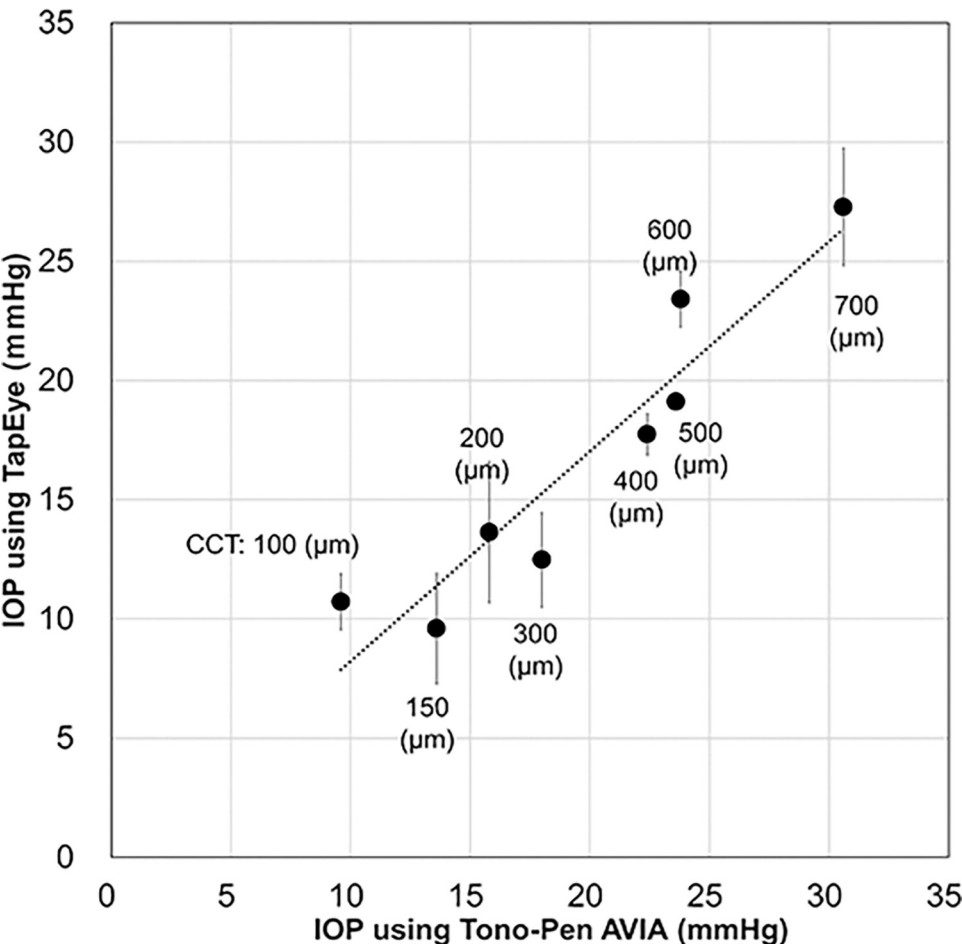

**Fig 4. Relationship of IOPs obtained from Tono-PenAVIA and TapEye tonometers using 3D-printed eyeball models with different CCTs.** The linear fit is plotted, and the correlation between the values is significant (Spearman r = 0.952, P<0.001). IOP = intraocular pressure; CCT = central corneal thickness.

**Table 1. Demographics of the participants.**

| Characteristic | Participants (n = 104) | |
|---|---|---|
| Age (yrs) | | |
| Mean ± SD | 40.3 ± 14.9 | |
| Range | 20 to 87 | |
| Sex (%) | | |
| Female | 60.6 | |
| | Right | Left |
| Central corneal thickness (μm) | | |
| Mean ± SD | 535.4 ± 38.4 | 534.6 ± 40.9 |
| Range | 408 to 619 | 399 to 619 |

SD = standard deviation

**Table 2. Comparison of IOP obtained using the TapEye, noncontact, and Goldmann applanation tonometers.**

| IOP (mmHg) | TET | c-TET | NCT | GAT | P value |
|---|---|---|---|---|---|
| Right eye at visit 1 (n = 104) | | | | | |
| Mean ± SD | 14.4 ± 2.5 | n/a | 16.1 ± 2.3 | 15.6 ± 2.6 | <0.001* |
| Range | 8.7 to 20.3 | | 11.1 to 20.8 | 10 to 21 | |
| Left eye at visit 1 (n = 104) | | | | | |
| Mean ± SD | 14.4 ± 2.2 | n/a | 16.1 ± 2.3 | 15.7 ± 2.5 | 0.013* |
| Range | 8.8 to 20.7 | | 9.2 to 21.4 | 10 to 22 | |
| Right eye at visit 2 (n = 101) | | | | | |
| Mean ± SD | 14.6 ± 2.1 | 16.0 ± 4.0 | 15.4 ± 2.3 | 15.6 ± 2.2 | <0.001† |
| Range | 9.4 to 20.6 | 5.0 to 27.1 | 5.0 to 20.9 | 10 to 21 | |
| Left eye at visit 2 (n = 101) | | | | | |
| Mean ± SD | 14.5 ± 2.0 | 16.0 ± 3.8 | 15.8 ± 2.4 | 15.6 ± 2.3 | <0.001† |
| Range | 9.0 to 19.3 | 1.6 to 26.1 | 7.0 to 20.6 | 10 to 21 | |

*Compared among the TET, NCT, and GAT data using one-way ANOVA.

†Compared among the c-TET, NCT, and GAT data using one-way ANOVA.

IOP = intraocular pressure; SD = standard deviation; TET = TapEye tonometer; NCT = noncontact tonometer; GAT = Goldmann applanation tonometer;

ANOVA = analysis of variance; n/a = not available.

participants were healthy with the exception of the presence of refractive errors. Of the 101 patients, 2 had nonprogressive keratoconus, 2 had undergone LASIK surgery more than 10 years prior, and 1 had normal-tension glaucoma in both eyes, which was treated with daily eye drops.

## Overall measurements

The mean TET, c-TET, NCT, and GAT IOP measurements are shown in **Table 2**. A comparison of the IOPs between the three devices revealed a significant difference in the right and left eyes at each visit using one-way ANOVA. At visit 1, the mean difference between the TET and GAT measurements was -1.3 ± 3.6 mmHg in the right eye and -1.4 ± 3.1 mmHg in the left eye. A significant difference was found between the TET and GAT measurements in each eye (P<0.001). At visit 2, the mean difference between the c-TET measurement and the GAT measurement was 0.4 ± 3.7 mmHg in the right eye and 0.5 ± 3.4 mmHg in the left eye. No significant differences were found between the c-TET and GAT measurements in each eye (P = 0.620 and 0.518, respectively).

## Association with Goldmann applanation tonometry

**Fig 5A–5D** shows scatterplots of the IOP measurements obtained by the TET and GAT at visits 1 and 2. The TET and GAT measurements were not normally distributed, Spearman's r correlation coefficient was low, and no significant correlation was found between the two tonometers. **Fig 5E and 5F** shows the scatterplots of the c-TET and GAT IOP measurements at visit 2. Spearman's r correlation coefficients were 0.408 and 0.414, and there was a significant correlation between the two tonometers (P<0.001). The IOPs between the NCT and GAT were significantly correlated at each visit in both the right and left eyes (**Table 3**).

**Fig 6** shows the distribution of agreement values between the c-TET and GAT measurements at visit 2 to assess the varying ranges of IOP. After calculating the c-TET values using visit 1 data, 50.5%, 84.2%, and 95.0% of eyes had agreement values of ± 2.5 mmHg, ± 5.0 mmHg, and ± 7.5 mmHg, respectively, between the c-TET and GAT measurements at visit 2.

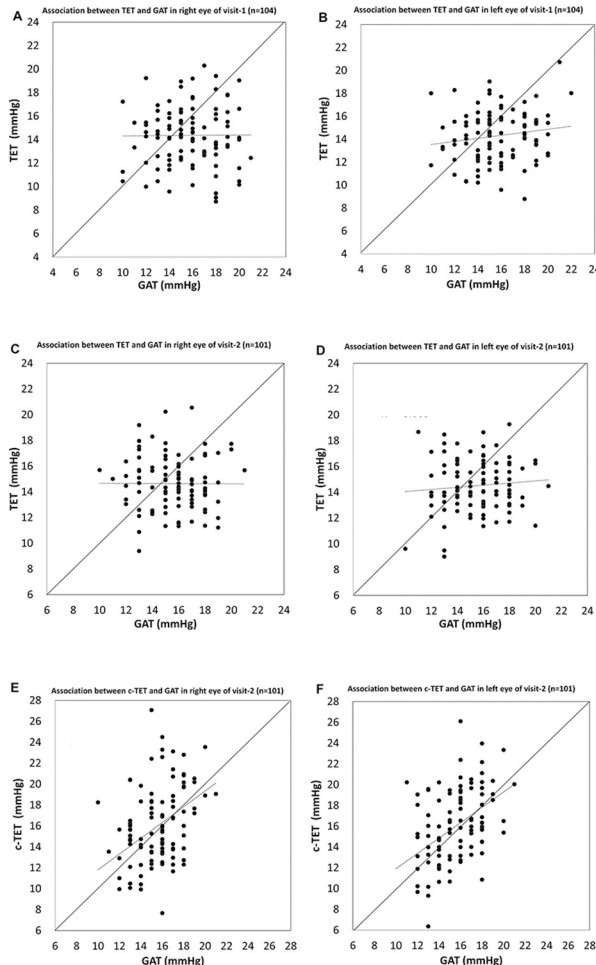

**Fig 5. Scatterplots showing the association between the IOP measured by the TET (c-TET) and GAT.** A, TET and GAT measurements in the right eye at visit 1. The linear fit was not significant (r = -0.005, P = 0.964; Spearman correlation coefficient). B, TET and GAT measurements in the left eye at visit 1. The linear fit was not significant (r = 0.140, P = 0.157). C, TET and GAT measurements in the right eye at visit 2. The linear fit was not significant (r = -0.046, P = 0.647). D, TET and GAT measurements in the left eye at visit 2. The linear fit was not significant (r = 0.074, P = 0.460). E, c-TET and GAT measurements in the right eye at visit 2. The linear fit was significant (r = 0.408, P<0.001). F, c-TET and GAT measurements in the left eye at visit 2. The linear fit was significant (r = 0.414, P<0.001). TET = TapEye tonometer; GAT = Goldmann applanation tonometer.

Bland–Altman analysis revealed a mean difference and 95% limits of agreement between the c-TET and GAT measurements (**Fig 7A and 7B**). Linear regression of the comparisons revealed a proportional error over the range of pressures examined. However, no significant

**Table 3. Correlation of IOPs obtained from the noncontact tonometer and Goldmann applanation tonometer.**

| Visit | Spearman correlation coefficient | P value |
|---|---|---|
| Right eye at visit 1 (n = 104) | 0.521 | <0.001 |
| Left eye at visit 1 (n = 104) | 0.427 | <0.001 |
| Right eye at visit 2 (n = 101) | 0.446 | <0.001 |
| Left eye at visit 2 (n = 101) | 0.530 | <0.001 |

IOP = intraocular pressure.

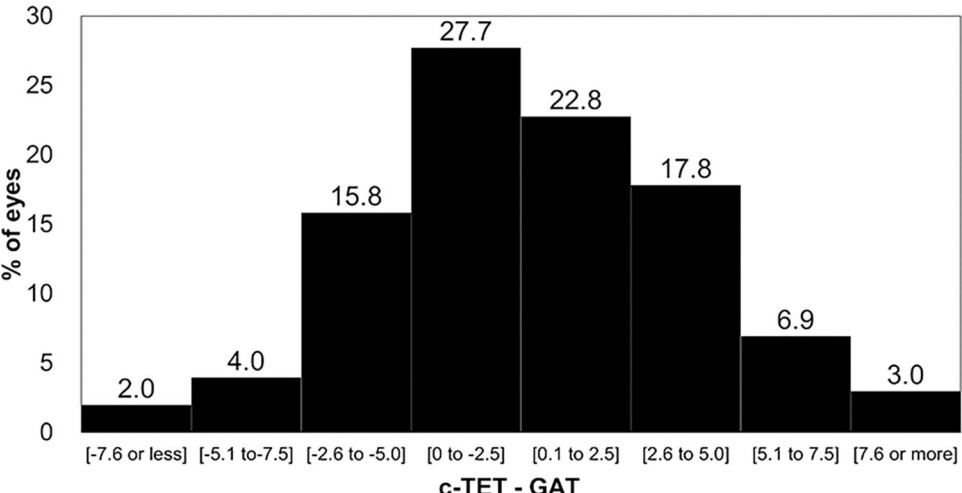

**Fig 6. Distribution of agreement between the c-TET and GAT measurements.** c-TET = corrected TapEye
tonometer; GAT = Goldmann applanation tonometer.

correlation was found between the values of c-TET-GAT and GAT at visit 2 in either the right
or left eye (**Fig 7C and 7D**).

No correlation was detected between central corneal thickness and any other parameters,
including the TET, c-TET, NCT, or GAT IOP measurement, or the difference between the c-
TET and GAT measurements.

## Repeatability

The repeatability of the TET was evaluated according to the ICC and CV using 2 sets of mea-
surements at visit 2 (**Table 4**). The mean ICC was less than 0.70, and the CV was less than 9%.

## Safety

Regarding adverse events, no vision-threatening complications were observed in this study,
indicating the safety of the three tonometers. Furthermore, no other adverse events, such as
abrasions, eyelid disorders, or conjunctivitis, occurred in any of the subjects at visits 1 or 2. No
significant differences in the slit-lamp microscopy findings were observed between visits 1 and
2.

## Discussion

In the current study, the IOP measured by the TET was not significantly correlated with that
measured by the GAT for either visit 1 or visit 2. The discrepancy between the two devices
might be attributed to two major reasons. First, the TET measurements may be affected by the
thickness and stiffness of the eyelids, which vary among individuals; this effect is particularly
pertinent to the transpalpebral approach. The thickness of the eyelid from the skin to the pal-
pebral conjunctiva is between 0.6 and 0.8 mm [12]. However, we did not evaluate the eyelid
thickness of each subject. To minimize the impact of the eyelid on the measurements, at visit
2, the corrected IOP of the TET was derived using the ΔIOP at visit 1. After implementing the
ΔIOP calculated at the initial measurement, the accuracy of the TET improved. It is not clear
whether the significant correlation between c-TET and GAT would be maintained for visit
intervals longer than 1 month. In terms of IOP self-monitoring, the accuracy of the TET

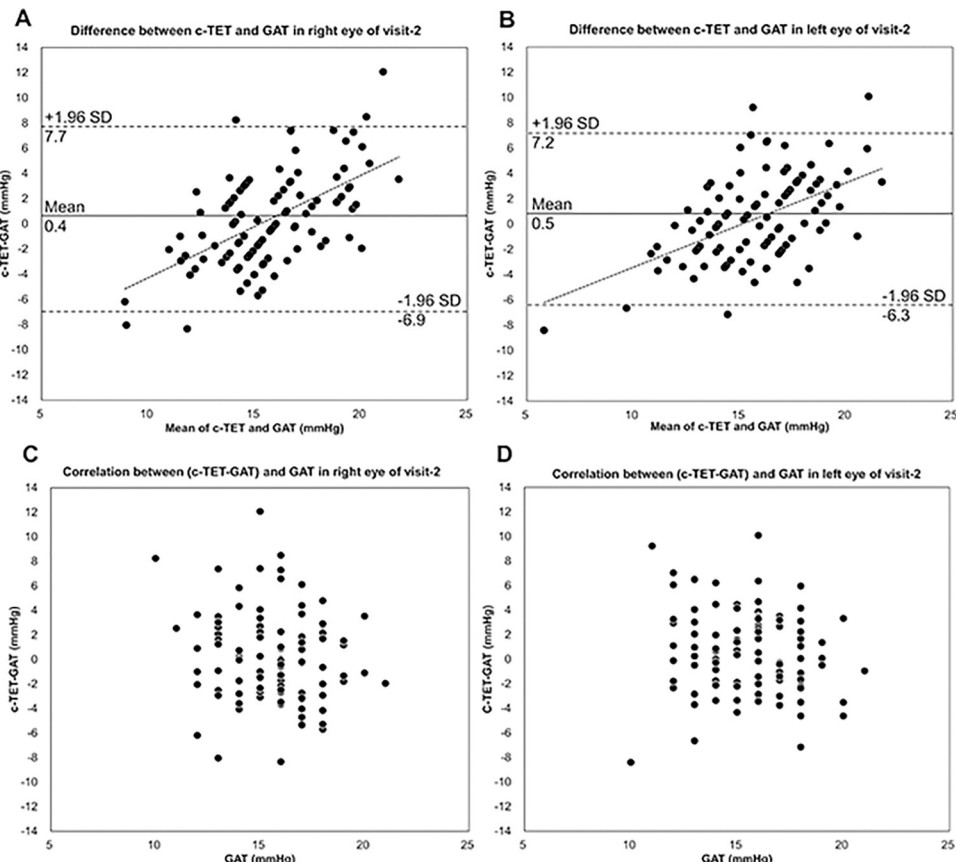

**Fig 7. Bland–Altman plot: A, Scatterplot showing the difference in IOP between corrected-TapEye tonometer (c-TET) and Goldmann applanation tonometer (GAT) measurements in the right eye.** The linear fit was significant (Spearman r = 0.530, P<0.001). B, Scatterplot showing the difference in the GAT and c-TET IOP measurements in the left eye. The linear fit was also significant (Spearman r = 0.451, P<0.01). Correlation plots: C, Scatterplot between the values of c-TET-GAT and GAT at visit 2 in the right eye (Spearman's r = -0.135, P = 0.178). D, Scatterplot between the values of c-TET-GAT and GAT at visit 2 in the left eye (Spearman's r = -0.192, P = 0.054).

measurements might not necessarily be important. In a clinical setting, we recommend that glaucoma patients start to use the TET after an initial visit in which both GAT and TET measurements are obtained to allow for future correction via the difference between the two. Our goal is to detect dynamic changes in IOP in patients with glaucoma or ocular hypertension in future work. If this is possible, the TET would demonstrate certain competitive advantages in terms of its ability to track relative changes in IOP and its safety. We are conducting a new study following the changes in IOP before and after the administration of glaucoma drops

**Table 4. Repeatability test for the intraocular pressure obtained with the TET at visit 2.**

| TET at visit-2 | Right eye (n = 101) | Left eye (n = 101) |
|---|---|---|
| ICC (95% CI) | 0.645 (0.473 to 0.761) | 0.698 (0.550 to 0.797) |
| CV (%) | | |
| Mean ± SD (range) | 8.2 ± 6.5 (0 to 39.0) | 7.5 ± 5.8 (0.2 to 24.6) |

ICC = intraclass correlation coefficient; CI = confidence interval; CV = coefficient of variation; SD = standard deviation.

using the TET and GAT. Second, the action of the subject pushing the button on the TET with their finger could also affect the IOP data. Individual differences in speed, alignment, and distance of pushing were observed despite having set appropriate thresholds for each in our tonometer software set. We are developing a new model of the TET with automatic motor driving to improve the stability and usability of the measurements.

In eyes with lower IOP, the TET measurements suggested that the accuracy of the present prototype was not adequate. The lowest measured c-TET value, 1.6, was obtained from a 75-year-old male without glaucoma and an IOP of 10 and 18.01 mmHg with GAT and TET, respectively, at visit 1 ($\Delta$IOP = 8.01). At visit 2, his IOP measured 10 and 9.62 mmHg with the GAT and TET, respectively. Thus, the c-TET was 1.61 mmHg. The difference in IOP might be due to the patient learning how to operate the device at visit 1 and/or the physical characteristics of the eyelid, which was loose and sagging. Since we need to be careful about $\Delta$IOP values in eyes with lower IOPs, the $\Delta$IOP might induce large errors in c-TET at visit 2. Two patients in our group had nonprogressive keratoconus, 2 patients had undergone LASIK, and 1 patient had normal-tension glaucoma treated with eye drops. We could not compare the IOPs obtained with TET for these patients with those obtained from normal subjects because of the small sample size. Generally, compared with GAT, corneal biomechanics measurements are recommended for patients with a history of corneal surgery and keratoconus when evaluating IOP.

The present study demonstrated that the c-TET IOP measurements were significantly correlated with the GAT measurements in subjects with normal IOPs. A total of 84.4% of participants were able to acquire IOP measurements using the TET within 5 mmHg of GAT measurements. With respect to rebound tonometers, Mudie et al. [13] reported that the Icare HOME and GAT measurements agreed within 5 mmHg in 91.3% of participants with glaucoma. Regarding transpalpebral tonometers, Sandner et al. [2] also reported that compared with GAT measurements, the IOP readings measured with the TGDc-01 were within an interval of 2 mmHg in 66.4% of participants, while 81.0% of the readings were within an interval of 3 mmHg. The mean difference between the TGDc-01 and GAT measurements was 0.71 ± 2.467 mmHg in their study. The superiority in terms of the measurement accuracy between the TET and TGDc-01 is unknown since the data have yet to be compared. The TGDc-01 is not useful for self-measurement by patients, as the examiner places the tip of the instrument on the margin of the patient's upper lid. Compared with the TGDc-01, the TET allows safe self-measurements after sufficient training and practice. In the research by Lam et al., the mean difference between the measurements obtained from a pressure phosphene tonometer and the GAT was -0.24 ± 1.57 mmHg; [14] between the two devices, 86% of the readings were within 2.0 mm Hg, and 91% were within 3.0 mm Hg [14]. Rai et al. [15] reported 3.5 ± 2.9 mmHg as the mean difference between GAT and pressure phosphene tonometer readings; they suggested that the regular use of a pressure phosphene tonometer significantly reduced patients' anxiety about their glaucoma. According to an overview of the previous literature, the IOP measurement accuracy of the Icare Home tonometer was greater than that of transpalpebral tonometers. Although our TET focuses on achieving the high accuracy exhibited by the Icare Home device, we emphasize that no disposable tips are needed for the operation, and patients need not be anxious about using the TET. Since the tip of the rebound tonometer is disposable, there is minimal risk of corneal infection and acrophobia. In the future, the TET has the potential to be an alternative to Icare Home if its usability and accuracy can be improved.

Since the correlations between the values of c-TET-GAT and GAT at visit 2 were not significant, our Bland–Altman analysis did not demonstrate consistency (Fig 7). Regarding the Icare device, the measurements obtained with the rebound tonometer underestimated the IOP with

respect to the GAT reading, whereas the measurements obtained with the TET showed an increasing overestimation of the IOP in comparison to the GAT measurement [16].

This pilot study has some limitations that should be considered. First, in this study, the measurements were limited to acquisition in the clinic because the TET is an off-label device and has not been approved by the relevant regulatory agencies. All measurements were performed by the participants except when the technicians were asked for instructions. The reliability of the TET depended on the user skill with the present prototype, resulting in the repeatability ranging from moderate to good. Better useability should be required to achieve a satisfactory product. Second, there were a few patients with hypertension or of elderly age. We are conducting another clinical trial assessing the IOPs measured by the TET in glaucoma patients with hypertension to determine whether similar results are obtained. Third, we did not quantify or qualify the thickness or stiffness of the upper eyelid, which might affect the IOP measured by the TET. To accurately measure the IOP with the TET, we used the corrected IOP measured at the second visit as much as possible. As there are individual differences in the thickness and stiffness of the upper eyelid, the TET is limited in terms of the accuracy of the measured IOP but easily assess fluctuations in this measurement. Some patients with glaucoma are administered prostaglandin analogs that easily induce deepening of the upper eyelid sulcus (DUES), which in turn causes the upper eyelid to become stiff and ptotic [17]. We need to compare the IOP using the TET between patients with and without DUES to evaluate the impact for abnormal eyelids. Fourth, as an inventor of the TET, Kobashi designed the research, collected the data, and performed analyses and interpretations in the present study. To reduce bias and improve confidence, we suggest that independent researchers conduct similar clinical studies and evaluate the efficacy and usability of the TET. When we move to the next phase, some of the independent researchers on our research team will be tasked with demonstrating clinical evidence of the advantages of the TET.

In conclusion, the results of our study show that the TET is safe to use, but the raw IOP data obtained by the device are not as reliable as those obtained by other self-tonometers. After calculating the difference between the TET and GAT measurements at the initial visit, the corrected IOP measured by the TET was correlated with the GAT at the second visit. The TET has the potential to address an unmet need by providing a tool for minimally invasive IOP measurements and therefore facilitating more informative data sharing between patients and physicians. Future research and development are needed to make our transpalpebral tonometer more useful for patients with glaucoma.

## Supporting information

**S1 Checklist. Reporting checklist for randomised trial.**
(PDF)

**S1 File.**
(PDF)

**S2 File.**
(PDF)

**S1 Video.**
(MP4)

## Author Contributions

**Conceptualization:** Hidenaga Kobashi.

**Data curation:** Hidenaga Kobashi.

**Formal analysis:** Hidenaga Kobashi.

**Funding acquisition:** Hidenaga Kobashi.

**Investigation:** Hidenaga Kobashi.

**Methodology:** Hidenaga Kobashi.

**Project administration:** Hidenaga Kobashi.

**Resources:** Hidenaga Kobashi.

**Software:** Hidenaga Kobashi.

**Supervision:** Hidenaga Kobashi.

**Validation:** Hidenaga Kobashi.

**Visualization:** Hidenaga Kobashi.

**Writing – original draft:** Hidenaga Kobashi.

**Writing – review & editing:** Hidenaga Kobashi.

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
