## [Decision Letter · Decision Letter 0]

18 Sep 2023

PONE-D-23-21962Evaluation of a new transpalpebral tonometer for self-measurement of intraocular pressurePLOS ONE

Dear Dr. Kobashi,

Thank you for submitting your manuscript to PLOS ONE. After careful consideration, we feel that it has merit but does not fully meet PLOS ONE’s publication criteria as it currently stands. Therefore, we invite you to submit a revised version of the manuscript that addresses the points raised during the review process.

We look forward to receiving your revised manuscript.

Kind regards,

Georgios Labiris, MD, PhD

Academic Editor

PLOS ONE

“This is based on results obtained from a project, JPNP0407002, subsidized by the New Energy and Industrial Technology Development Organization (NEDO).”

“H.K.: CEO and equity owner, Toneasy Inc.; Patent, Toneasy Inc.”

Reviewers' comments:

Reviewer's Responses to Questions

**Comments to the Author**

1. Is the manuscript technically sound, and do the data support the conclusions?

Reviewer #1: Partly

Reviewer #2: Yes

2. Has the statistical analysis been performed appropriately and rigorously? 

Reviewer #1: N/A

Reviewer #2: Yes

3. Have the authors made all data underlying the findings in their manuscript fully available?

Reviewer #1: No

Reviewer #2: Yes

4. Is the manuscript presented in an intelligible fashion and written in standard English?

Reviewer #1: Yes

Reviewer #2: Yes

5. Review Comments to the Author

Reviewer #1: The manuscript by Kobashi describes TapEye tonometer (TET), a novel transpalpebral self-tonometer, to obtain intraocular pressure (IOP) measurement based on palpation of the upper eyelid. They found that the corrected-IOP value measured by the TET (c-TET), calculated based on the difference between the TET and GAT measurements at the first visit, was correlated with GAT measurements. The TET has the potential to meet the need for minimally invasive IOP measurements. Although the topic area is potentially interesting, but I do have some concerns regarding the level of experimental detail provided and a lack of clarity in the presentation and description of the data that makes it difficult to establish conclusions from the study.

1. Firstly, the author measure the IOPs of several 3D-printed eyeball models with various central corneal thicknesses using TET and Tono-Pen AVIA tonometers. Do these models have just eyeballs, or do they also have eyelids?

2. Could the level of stiffness of the eyelids affect the measurement results and how to correct that part?

3. With only one glaucoma patient among the participants, how accurate is TET performance in individuals with low or high intraocular pressure? With the lowest measured value being 1.6 in the c-TET group and 10 in the GAT group, the deviation is substantial. Does this indicate inadequate accuracy of TET measurements in cases of low intraocular pressure?

4. The subjects can seek help or receive instructions from technicians when using TET in both visits. How do they perform if there is no access to TET instructions? Are the results still reliable?

5. How many times of measurements are needed to obtain a TET IOP value which successfully pass the criteria? In a single measurement session, what range of differences between the data points is considered acceptable? Do the failure measurements affect the IOP value before a successful TET measurement is obtained?

6. The significant correlation between TET and GAT seems to rely on correction based on the difference between the TET and GAT measurements in visit 1. In this study, the visit interval is one month. Is the correlation between c-TET and GAT still significant if the visit interval is prolonged?

7. It is mentioned that no vision-threatening complications were observed in this study. What is the incidence of abrasions, eyelid disorder, and conjunctivitis in this study?

8. Patients with a history of incisional glaucoma surgery, LASIK, corneal surgery, or history of previously diagnosed keratoconus were not excluded in this study. How does the TET perform when measuring these patients? Is the GAT sufficiently accurate as gold standard considering these patients?

9. In line 287, the author claimed that our TET can detect the dynamic change in IOP in patients with glaucoma or ocular hypertension. However, the participants’ IOP measured by GAT all falls within the range of 22 or below. Insufficient information is available to draw the conclusion mentioned above.

10. Of eyelid tonometer TGDc-01 IOP values, 66.4% (132 eyes) within ±2 mmHg and 81.0% (161 eyes) within ±3 mmHg of those with applanation tonometry, and 10.0% (20 eyes) the deviation from applanation tonometry was more than 4 mmHg. Only 50.5% of c-TET measurements were within 2.5 mmHg of GAT measurements. What are the pros and cons of TET compared with TGDc-01?

11. What are the factors related to difference between the TET and GAT measurements? Do eyelid thickness or central corneal thickness affect difference between the TET and GAT measurements?

12. In figure 6, the label corresponding to the left fourth column may be incorrect.

13. As Kobashi designed the research, collected data, and also analysis and interpretation, How could you reduce bias and improve confidence?

Reviewer #2: It tis a interesting device for intraocular pressure measurement. There are some opinios：

1）When making aTET or GAT measurement, whether it is measured once or three times to take the average value？

2）How to consider the factors of the eyelid, whether the single eyelid or double eyelid is considered when selecting a patient？

3）The abscissa in Figure 7 is mean of c-TET and GAT, but in Figure 5 it is GAT. This is unreasonable. If Fig7 is a Bland-Altmen plot to illustrate consistency, you can provide an additional correlation plot with GAT as abscissa.

4）In Figure 7，although most of the points fall between 95% limits of agreement, the deviation from the upper and lower limits actually far exceeds the clinical approval value

5）In the study, the NCT test method was also used. It is recommended to provide the consistency analysis results of NCT and GAT as a reference to clarify the equipment needs to be improved.

6. PLOS authors have the option to publish the peer review history of their article (what does this mean?). If published, this will include your full peer review and any attached files.

Reviewer #1: No

Reviewer #2: **Yes: **Xu Chen

---

## [Author Response · Author response to Decision Letter 0]

20 Dec 2023

Dr. Georgios Labiris,

Thank you for your e-mail regarding our manuscript (PONE-D-23-21962) titled “Evaluation of a new transpalpebral tonometer for self-measurement of intraocular pressure” as well as for the comments from the reviewer. We believe that the paper has been much improved, largely due to the referees’ many thoughtful comments. We would like to respond below to each comment.

To Editorial Office:

Thank you.

“This is based on results obtained from a project, JPNP0407002, subsidized by the New Energy and Industrial Technology Development Organization (NEDO).”

We obtained funding as JPY 24,770,681 in this grant by NEDO.

The information on NEDO is shown in the following website:

https://www.nedo.go.jp/english/introducing/introducing_index.html

[Page 1, lines 9-13]: One sentences has been added to the TITLE page.

”

The statement has been added as above.

The statement has been added as above.

The statement has been added as above.

The statement has been added as above.

“H.K.: CEO and equity owner, Toneasy Inc.; Patent, Toneasy Inc.”

The statement has been added to the revision and cover letter.

Thank you for your comment. I have deleted the “data not shown” and revised the manuscript.

[Page 21, lines 361-362]: One sentence has been deleted.

“We are also developing an iPhone app that can communicate with the TET (data not shown).”

To Reviewer #1:

The manuscript by Kobashi describes TapEye tonometer (TET), a novel transpalpebral self-tonometer, to obtain intraocular pressure (IOP) measurement based on palpation of the upper eyelid. They found that the corrected-IOP value measured by the TET (c-TET), calculated based on the difference between the TET and GAT measurements at the first visit, was correlated with GAT measurements. The TET has the potential to meet the need for minimally invasive IOP measurements. Although the topic area is potentially interesting, but I do have some concerns regarding the level of experimental detail provided and a lack of clarity in the presentation and description of the data that makes it difficult to establish conclusions from the study.

Thank you for your review and positive comments. I have clarified the experimental details and modified the conclusions.

1. Firstly, the author measure the IOPs of several 3D-printed eyeball models with various central corneal thicknesses using TET and Tono-Pen AVIA tonometers. Do these models have just eyeballs, or do they also have eyelids?

Our 3-D printed eyeball model does not include an eyelid. To simplify the simulation of the elevated IOP model, we used artificial eye models without eyelids by changing the corneal thickness based on a 3D printing approach.

[Page 8, lines 95-97]: One sentence has been added.

“To simplify the simulation of the elevated IOP model, we used artificial eye models without eyelids by changing the corneal thickness based on a 3D printing approach.”

2. Could the level of stiffness of the eyelids affect the measurement results and how to correct that part?

As mentioned in the Discussion, we did not evaluate the stiffness of the eyelids, which might affect the IOP measurement obtained by the TET. To achieve accurate IOP measurement with the TET, we used the corrected-IOP value measured at the second visit as much as possible. As individual differences in thickness and stiffness of the upper eyelid occur, the TET is limited in terms of the accuracy of the measured IOP but has the advantage of easy assessment of IOP fluctuations. 

[Page 22, lines 384-388]: Two sentences have been added.

“To accurately the IOP of the TET, we used the corrected-IOP value measured at the second visit as much as possible. As there are individual differences in thickness and stiffness of the upper eyelid, the TET is limited in terms of its accuracy in measuring the IOP, but is advantageous in allowing easy assessment of IOP fluctuations.”

3. With only one glaucoma patient among the participants, how accurate is TET performance in individuals with low or high intraocular pressure? With the lowest measured value being 1.6 in the c-TET group and 10 in the GAT group, the deviation is substantial. Does this indicate inadequate accuracy of TET measurements in cases of low intraocular pressure?

Thank you for your advice. As you mentioned, the TET measurements did not indicate the suitable accuracy in eyes with lower IOP of the present prototype. Indeed, the example with the lowest measured c-TET value of 1.6 is described as follows. A 75 years old male without glaucoma had an IOP of 10 and 18.01 mmHg with the GAT and TET at visit 1, respectively (ΔIOP=8.01). At visit 2, his IOP measured 10 and 9.62 mmHg with the GAT and TET, respectively. Thus, c-TET was 1.61 mmHg. The reason for the difference in IOP might be due to the patient learning how to operate the device at visit 1 and his sagging eyelid. Since we need to be careful about ΔIOP in eyes with lower IOPs, ΔIOP might induce large errors in c-TET at visit 2.

[Page 20, lines 321-328]: Seven sentences have been added.

“In eyes with lower IOP, TET measurements did not indicate adequate accuracy of the present prototype. Indeed, the example with the lowest measured c-TET value of 1.6 is described as follows. A 75 years old male without glaucoma had an IOP of 10 and 18.01 mmHg with the GAT and TET at visit 1, respectively (ΔIOP=8.01). At visit 2, his IOP measured 10 and 9.62 mmHg with the GAT and TET, respectively. Thus, c-TET was 1.61 mmHg. The reason for the difference in IOP might be due to the patient learning how to operate the device at visit 1 and his sagging eyelid. Since we need to be careful about ΔIOP in eyes with lower IOPs, ΔIOP might induce large errors in c-TET at visit 2.”

4. The subjects can seek help or receive instructions from technicians when using TET in both visits. How do they perform if there is no access to TET instructions? Are the results still reliable?

When the subjects asked for help or held the device in an inappropriate manner and pushed the button, the technicians gave the proper instructions while standing by their side at both visits. The reliability of the TET depended on the user skill in the present prototype, as the repeatability showed moderate to good results.

[Page 11, lines 162-164]: One sentence has been modified.

“When the subjects asked for help or held the device in an inappropriate manner and pushed the button, the technicians gave the proper instructions while standing by their side at both visits.”

[Page 22, lines 378-389]: One sentence has been added.

“The reliability of TET depended on the user skill with the present prototype, as the repeatability showed moderate to good results.”

5. How many times of measurements are needed to obtain a TET IOP value which successfully pass the criteria? In a single measurement session, what range of differences between the data points is considered acceptable? Do the failure measurements affect the IOP value before a successful TET measurement is obtained?

At least seven successful measurements were required to obtain the IOP value with the TET. The average value of seven measurements was considered the IOP value for the TET. Failed measurements were excluded by the IOP calculation.

[Page 10, lines 149-151]: Two sentences have been added.

“The average value of the seven measurements was considered the IOP value of the TET. Failed measurements were excluded by the IOP calculation.”

6. The significant correlation between TET and GAT seems to rely on correction based on the difference between the TET and GAT measurements in visit 1. In this study, the visit interval is one month. Is the correlation between c-TET and GAT still significant if the visit interval is prolonged?

It is not clear whether the significant correlation between c-TET and GAT would be maintained for visit intervals longer than 1 month. To reduce the difference between the TET and GAT measurements, we focused on the ΔIOP at visit 1 and optimized c-TET at visit 2.

[Page 19, lines 305-307]: One sentence has been added.

“It is not clear whether the significant correlation between c-TET and GAT would be maintained for visit intervals longer than 1 month.”

7. It is mentioned that no vision-threatening complications were observed in this study. What is the incidence of abrasions, eyelid disorder, and conjunctivitis in this study?

Other adverse events, such as abrasions, eyelid disorder, and conjunctivitis, did not occur in any subject at visits 1 and 2.

[Page 19, lines 292-293]: One sentence has been added.

“Other adverse events such as abrasions, eyelid disorder, and conjunctivitis did not occur in any subject at visit 1 and 2.”

8. Patients with a history of incisional glaucoma surgery, LASIK, corneal surgery, or history of previously diagnosed keratoconus were not excluded in this study. How does the TET perform when measuring these patients? Is the GAT sufficiently accurate as gold standard considering these patients?

As you mentioned, patients with a history of incisional glaucoma surgery, LASIK, or corneal surgery were not excluded, nor were patients who had undergone cataract extraction at least one month prior. We did not exclude patients with a history of previously diagnosed keratoconus. However, there were only 2 patients with nonprogressive keratoconus, 2 patients with LASIK, and 1 patient with normal-tension glaucoma treated with eye drops. We could not compare the IOPs obtained with the TET of these patients with those of normal subjects because of the small sample size. Generally, compared with those of the GAT, corneal biomechanics measurements are recommended in patients with a history of corneal surgery and keratoconus in terms of the evaluation of IOP.

[Page 20, lines 328-334]: Three sentences have been added.

“Only 2 patients in our group had nonprogressive keratoconus, 2 patients had undergone LASIK, and 1 patient had normal-tension glaucoma treated with eye drops. We could not compare the IOPs obtained with TET for these patients with those obtained from normal subjects because of the small sample sizes. Generally, compared with GAT, corneal biomechanics measurements are recommended in patients with a history of corneal surgery and keratoconus in terms of the evaluation of the IOP.”

9. In line 287, the author claimed that our TET can detect the dynamic change in IOP in patients with glaucoma or ocular hypertension. However, the participants’ IOP measured by GAT all falls within the range of 22 or below. Insufficient information is available to draw the conclusion mentioned above.

This sentence has been deleted because of its exaggerated tone. Our goal is to detect the dynamic change in IOP in patients with glaucoma or ocular hypertension in future work.

[Page 19, lines 312-314]: One sentence has been added.

“Our goal is to detect the dynamic changes in the IOP in patients with glaucoma or ocular hypertension in future work.”

[Page 19, lines 310-312]: One sentence has been deleted.

“Similar to the use of an individually owned blood pressure monitor in daily life, our TET can detect the dynamic change in IOP in patients with glaucoma or ocular hypertension.”

10. Of eyelid tonometer TGDc-01 IOP values, 66.4% (132 eyes) within ±2 mmHg and 81.0% (161 eyes) within ±3 mmHg of those with applanation tonometry, and 10.0% (20 eyes) the deviation from applanation tonometry was more than 4 mmHg. Only 50.5% of c-TET measurements were within 2.5 mmHg of GAT measurements. What are the pros and cons of TET compared with TGDc-01?

We cannot speak of the superiority of the measurement accuracy between TET and TGDc-01 because the data between the two have yet to be compared. The TGDc-01 is not useful as an instrument for self-measurement, as an examiner needs to place the tip of the instrument on the patient’s upper lid margin. Compared with the TGDc-01, the TET has the advantage of self-measurement safely after sufficient training and practice.

[Page 21, lines 343-348]: Three sentences have been added.

“The superiority in terms of the measurement accuracy between the TET and TGDc-01 is unknown since, to date, the data have yet to be compared. The TGDc-01 is not useful as an instrument for self-measurement by patients, as the examiner places the tip of the instrument on the patient’s upper lid margin. Compared with the TGDc-01, the TET has the advantage of allowing self-measurement safely after sufficient training and practice.”

11. What are the factors related to difference between the TET and GAT measurements? Do eyelid thickness or central corneal thickness affect difference between the TET and GAT measurements?

No significant factors were detected when we looked for an association with ΔIOP (TET-GAT) in the present study. The central corneal thickness was not significantly associated with ΔIOP. Unfortunately, we did not evaluate the eyelid thickness in each subject. As individual differences in thickness and stiffness of the upper eyelid occur, TET has a limitation in terms of accuracy of IOP.

[Page 22, lines 386-388]: One sentence has been added.

“As there are individual differences in thickness and stiffness of the upper eyelid, the TET is limited in terms of the accuracy of the measured IOP but has the advantage of easy assessment of IOP flu

---

## [Decision Letter · Decision Letter 1]

19 Feb 2024

PONE-D-23-21962R1Evaluation of a new transpalpebral tonometer for self-measurement of intraocular pressurePLOS ONE

Dear Dr. Kobashi,

Thank you for submitting your manuscript to PLOS ONE. After careful consideration, we feel that it has merit but does not fully meet PLOS ONE’s publication criteria as it currently stands. Therefore, we invite you to submit a revised version of the manuscript that addresses the points raised during the review process.

We look forward to receiving your revised manuscript.

Kind regards,

Georgios Labiris, MD, PhD

Academic Editor

PLOS ONE

Journal Requirements:

Reviewers' comments:

Reviewer's Responses to Questions

**Comments to the Author**

1. If the authors have adequately addressed your comments raised in a previous round of review and you feel that this manuscript is now acceptable for publication, you may indicate that here to bypass the “Comments to the Author” section, enter your conflict of interest statement in the “Confidential to Editor” section, and submit your "Accept" recommendation.

Reviewer #3: (No Response)

Reviewer #4: All comments have been addressed

2. Is the manuscript technically sound, and do the data support the conclusions?

Reviewer #3: Partly

Reviewer #4: (No Response)

3. Has the statistical analysis been performed appropriately and rigorously? 

Reviewer #3: Yes

Reviewer #4: (No Response)

4. Have the authors made all data underlying the findings in their manuscript fully available?

Reviewer #3: Yes

Reviewer #4: (No Response)

5. Is the manuscript presented in an intelligible fashion and written in standard English?

Reviewer #3: No

Reviewer #4: (No Response)

6. Review Comments to the Author

Reviewer #3: Dear Editor,

Thank you for the opportunity to provide a review of Manuscript PONE-D-23-21962 entitled "Evaluation of a new transpalpebral tonometer for self-measurement of intraocular pressure." My comments relate primarily to the adequacy of the implementation and reporting of epidemiologic and statistical procedures.

I understand that this manuscript underwent a previous round of peer review. I have reviewed both the original and revised versions of the manuscript, including the author's response to the first-round comments.

The quality of the technical English is appropriate but variable. While offering no bar to my evaluation of the manuscript, many errors were present. I strongly recommend that the authors conduct a thorough round of copyediting to eliminate the grammatical and syntactical errors in the text.

# Reporting of correlation results

The authors state that they will report Pearson's correlation coefficient (or Spearman's). However, the authors do not report the coefficient of correlation (r) in any of the scatterplots provided, but instead report the coefficient of determination (r^2). (See Figure 4, all graphs in Figure 5, and all graphs in Figure 6.)

The authors need to clarify whether this is a labelling or calculation error. That is, did they actually calculate r^2 or did they simply label the r results as r^2. It is imperative that they DO NOT present r^2 results because the meaning of the coefficient of determination is very different from the coefficient of correlation. This is a major error that requires careful attention.

# Description of the device and the process of use

Still images are provided by the authors to describe the device and demonstrate its usage. This is substandard. The journal allows for the submission of video clips, and this is one time when such a submission is preferable to the textual and still image descriptions of the device and its use.

# Consistency of presentation of means and standard deviations

The authors state in the methods that they will present outcomes as means plus-minus standard deviations. The authors inconsistently carry this approach throughout the text. In many places, the authors report the mean (SD). The authors need to observe consistency in their reporting.

# Recommendation

The issues identified above are relatively manageable, but I cannot support the acceptance of the manuscript until they are addressed.

Thank you.

Reviewer #4: (No Response)

7. PLOS authors have the option to publish the peer review history of their article (what does this mean?). If published, this will include your full peer review and any attached files.

Reviewer #3: No

Reviewer #4: No

---

## [Author Response · Author response to Decision Letter 1]

29 Feb 2024

Dr. Georgios Labiris,

Thank you for your e-mail regarding our manuscript (PONE-D-23-21962R1), titled “Evaluation of a new transpalpebral tonometer for self-measuring intraocular pressure”, as well as for the comments from the reviewers. We believe that the paper has been greatly improved, largely due to the referees’ many thoughtful comments. We would like to respond below to each comment.

To Reviewer #3:

Thank you for the opportunity to provide a review of Manuscript PONE-D-23-21962 entitled "Evaluation of a new transpalpebral tonometer for self-measurement of intraocular pressure." My comments relate primarily to the adequacy of the implementation and reporting of epidemiologic and statistical procedures.

Thank you for your review and positive comments. I have clarified the experimental representations in this revision.

I understand that this manuscript underwent a previous round of peer review. I have reviewed both the original and revised versions of the manuscript, including the author's response to the first-round comments.

Modifications have been made based on the comments of all the reviewers.

The quality of the technical English is appropriate but variable. While offering no bar to my evaluation of the manuscript, many errors were present. I strongly recommend that the authors conduct a thorough round of copyediting to eliminate the grammatical and syntactical errors in the text.

To eliminate grammatical and syntactical errors in the text, this revision was examined by a peer review of native English experts.

# Reporting of correlation results

The authors state that they will report Pearson's correlation coefficient (or Spearman's). However, the authors do not report the coefficient of correlation (r) in any of the scatterplots provided, but instead report the coefficient of determination (r^2). (See Figure 4, all graphs in Figure 5, and all graphs in Figure 6.)

The authors need to clarify whether this is a labelling or calculation error. That is, did they actually calculate r^2 or did they simply label the r results as r^2. It is imperative that they DO NOT present r^2 results because the meaning of the coefficient of determination is very different from the coefficient of correlation. This is a major error that requires careful attention.

I apologize for the confusion. The coefficient of determination (R2) was deleted in each figure. Spearman's correlation coefficients are now reported in the figure legends.

# Description of the device and the process of use

Still images are provided by the authors to describe the device and demonstrate its usage. This is substandard. The journal allows for the submission of video clips, and this is one time when such a submission is preferable to the textual and still image descriptions of the device and its use.

I have added a video illustrating the device and the process of its use as a supplemental video.

# Consistency of presentation of means and standard deviations

The authors state in the methods that they will present outcomes as means plus-minus standard deviations. The authors inconsistently carry this approach throughout the text. In many places, the authors report the mean (SD). The authors need to observe consistency in their reporting.

The outcomes are presented as the means ± standard deviations in this revision.

# Recommendation

The issues identified above are relatively manageable, but I cannot support the acceptance of the manuscript until they are addressed.

The above issues have been addressed.

We believe that the manuscript has been satisfactorily prepared and submitted, and we hope that it will be accepted for publication in PLoS ONE. Thank you for your attention and consideration.

Sincerely yours,

Hidenaga Kobashi, MD, PhD, Department of Ophthalmology, Keio University, School of Medicine, Tokyo, Japan. E-mail address: hidenaga_kobashi@keio.jp

---

## [Editor Report · Decision Letter 2]

8 Apr 2024

Evaluation of a new transpalpebral tonometer for self-measuring intraocular pressure

PONE-D-23-21962R2

Dear Dr. Kobashi,

We’re pleased to inform you that your manuscript has been judged scientifically suitable for publication and will be formally accepted for publication once it meets all outstanding technical requirements.

Kind regards,

Georgios Labiris, MD, PhD

Academic Editor

PLOS ONE

---

## [Editor Report · Acceptance letter]

26 Apr 2024

PONE-D-23-21962R2 

PLOS ONE

Dear Dr. Kobashi, 

I'm pleased to inform you that your manuscript has been deemed suitable for publication in PLOS ONE. Congratulations! Your manuscript is now being handed over to our production team.

Kind regards, 

on behalf of

Dr. Georgios Labiris 

Academic Editor

PLOS ONE